# The Practices and Positionings of a Postcolonial Counterpublic: An Analysis of Black Lives Matter in Denmark

## Bolette B. Blaagaard

Department of Communication and Psychology, Aalborg University, 2450 Copenhagen, Denmark; blaagaard@ikp.aau.dk

**Abstract:** Drawing on postcolonial critique to analyze the work and political purpose of activist groups on social media, this article asks the question: How do digital media communications simultaneously reinstate binary oppositions and invite rhizomatic relations? While the concept of counterpublics is helpful when it comes to understanding the voices of opposition in public discourse, it is also necessary to introduce postcolonial critique and geopolitical and historical distinctions in order to grasp the particularities of global digital activism (Brouwer and Paulesc 2017; Blaagaard 2018). This article does exactly that: Illustrating the postcolonial, hybrid, and cosmopolitan qualities of digital activism on social media platforms, the article presents a discursive analysis of Black Lives Matter Denmark (BLM-DK) as they operate on the social media platform Facebook. The group's posts are dedicated to juridical and political struggles over discrimination and racial violence in Denmark and the United States, thus producing a counterpublic. The posts moreover introduce and connect two very different geopolitical and historical contexts, thus showing social media's potential for creating rhizomatic relations.

**Keywords:** BLM-DK; postcolonial critique; counterpublics; digital activism; connective media





## 1. Introduction

Black Lives Matter (BLM) is one of the most visible and respected social movements using digital media to organize and mobilize support for reforms of the US justice and prison system and for granting greater equality to the African American population in the United States (US). Built on the continuous struggle for civil rights, the activist social movement is arguably quintessentially American. However, BLM has chapters across Europe (Milman et al. 2021), which implicates Europe's colonial past and political present in the organization's narrative and aims. While the early civil rights struggle in the US is associated with physical sit-ins and marches in person, the current BLM organization's systematic use of digital social media in the postcolonial space of Europe calls for a rethinking and reconceptualization of the relationship between digitalization and postcoloniality and the concomitant rhizomatic structures of transnational activism. Media-centric theories of technologically organized social movements fall short of producing an adequate understanding of digital activism in postcolonial societies. In this article, I focus on the digital work of Black Lives Matter Denmark (BLM-DK) to explore what a digital and postcolonial activist position may look like and what implications may follow from the digitalization, showing how the visual, linguistic, and, not least, technological discourses on BLM-DK's Facebook page reinstate binary oppositions as well as invite rhizomatic relations.

Postcolonial critique and media studies have long had a complicated relationship (Hall 1995; Grossberg 2002; Shome 2016; Blaagaard 2020). From a postcolonial perspective, media have tended to reproduce the power structures and binaries underlying Western epistemologies. However, media technologies have developed from affirming the Western binary structures of "'here' and 'there', 'then' and 'now', 'home' and 'abroad' perspectives" (Hall 1995, p. 251) to a tool for potentially diversifying public participation and perspectives.

Meanwhile, postcolonial critique has progressed from rather Eurocentric literary and comparative studies of deconstruction while cultural and political studies of decolonization and critiques of marketization and potential exploitation of affective labor online are being challenged by a young generation of digital entrepreneurs (Steele 2021; see also Ponzanesi 2014). On the one hand, digital technologies and new media help counterpublics to emerge and empower minorities and oppressed communities, such as in the example of BLM. Counterpublics, as the concept is developed and discussed by theorists such as Fraser (1991), Squires (2002), and Warner (2002), are discursive spaces of counternarratives and political opposition to the dominant public. On the other hand, the idea of counterpublics itself runs the risk of reinstating the binary pairs inherent in Western epistemologies and allows the media industry and market to take advantage of vulnerable people by exploiting their digital creativity and marketing them as 'exotic' or 'ethnic' inputs into an otherwise Westernized dominant public. While the concept of counterpublics is helpful when it comes to understanding the voices of opposition in public discourse and how they produce a space of empowerment for minoritized subjects, it is also necessary to introduce postcolonial critique and geopolitical and historical distinctions in order to grasp the particularities of global digital activism (Brouwer and Paulesc 2017; Blaagaard 2018). In the case of BLM-DK, which was formed by activist Bwalya Sørensen in 2016, the Danish historical and political context is meshed with the historical and political context of the US while simultaneously producing distinct and varied narratives and expressions. Understanding how postcolonial vernacular expressions and practices utilize and counteract the marketization of new media initiates an approach that may shed light on the complex nature of the relationship between media and the postcolonial. Navigating these shifting waters of technology and postcolonial critique warrants a sensitivity towards the embodiedness of digital actors and their historically and socially embedded circumstances. This article, then, argues that a digital perspective on social media activism is limited in scope and should be critically discussed using postcolonial theories and practices to understand the nuances of activist work.

## 2. A Note on Method

I present a small sample of social media posts gathered manually from BLM-DK's Facebook page between 4 March 2022 and 6 May 2022, both dates included. Facebook is chosen because it is the social media platform which is used by the organization. The months of sampling are randomly selected for the purpose of discussing digital activism and postcolonial critique. BLM-DK is, then, a critical case study which allows for the logical deduction that what is true in this case is true for other cases (Flyvbjerg 1991, p. 475) of European, postcolonial, digitally organized social movements. Within the selected timeframe, BLM-DK posts 24 posts containing links to activism in four different countries, the United Kingdom (UK), the United States (US), Spain, and Denmark, and shares posts from at least 17 different media outlets, platforms, and personalities[1]. Despite the limited size of the sample, it includes a surprisingly rich and broad variety of expressions and presents a range of issues, which allow me to explore what a digital and postcolonial counterpublic may look like and how it produces binary oppositions as well as rhizomatic relations—beyond the technological imperative. Due to the starting point of the sample and the geographical situatedness of BLM-DK, many of the posts deal with the implications of the Russian invasion of Ukraine for Black and Brown people at the border between Ukraine and Poland. Danish policies regarding refugees and migrants at the time are also heavily featured. While these issues are temporally located, they point to broader issues in postcolonial societies that will be discussed.

I analyze the posts using a discursive reading inspired by Stuart Hall (1995, 2002) and Rose Gillian (2016), identifying firstly the postcolonial *positionings* negotiated through the posts by BLM-DK, followed by a discussion of the postcolonial *practices* involved. I employ linguistic and visual discursive analyses and aim to illustrate the changing same of postcolonial expressions on social media using the theoretical framework of the

practice concepts of *vernacular expressions*, *embodied hashtags*, and *survival technology* and the positioning concepts of *border thinking* and *cosmopolitan bridges* to understand their enabling of the postcolonial counterpublic.

To present and discuss this conceptual framework, in the following, I begin by presenting an understanding of digital activism, which takes as a starting point the technologies underlying the activist practices (the activists of new media). I then move on to presenting an understanding of postcolonial activism to explore historically and socially embedded practices. I present the analytical framework made up of the concepts of *vernacular expressions*, *embodied hashtags*, *survival technologies, border thinking*, and *cosmopolitan bridges* (the postcolonial activist), which are all concepts teased out of postcolonial theories and discussions. I bring the theoretical framework to bear on the case study of BLM-DK to illustrate the postcolonial qualities of digital activism on social media platforms, producing a discursive analysis of Black Lives Matter Denmark (BLM-DK) as they operate on the social media platform Facebook (the positionings and bridges of Black Lives Matter Denmark (BLM-DK) and postcolonial and vernacular practices). Despite the US starting point, the organization's posts are dedicated to juridical and political struggles over discrimination and racial violence in Denmark and beyond, introducing and connecting several very different geopolitical and historical contexts while utilizing the corporate structures of Facebook to do so.

### 3. The Activists of New Media

New digital media have changed the way activism is mobilized: from collective politics of change to a connective hydra of causes claimed and pursued by a variety of stakeholders enabled and made visible by digital media networks (Bennett and Segerberg 2012; Fenton 2012; Blaagaard and Roslyng 2022).[2] The early civil rights struggle of the sixties is, to Danes, markedly American, whereas the digital activism of BLM produces both transnational solidarity and local political right claims. Bennett and Segerberg (2012) theorize the concept of connective action, which is generated by new media's ability to connect networks of political actors. While the authors describe and define the organization of digital activism as falling into two logics, the logics of collective and connective action (p. 748), they relegate collective action to an affective and embodied practice which promotes contestation through common identification and cause, thus "inviting personalized interpretations of problems and self-organization of action" (p. 755). Social movement organizations, such as the civil rights movements of the sixties, for instance, offer "straightforward membership on the basis of gender, religion or life cycle" and are controlled by leaders who present the ideology of the movement to the press and other stakeholders (Milan 2015, p. 59). Identity- and memory-based collective action embeds activists in a particular embodied counterpublic fighting against historical, social, and political structures as well as trying to recast those structures in ways that recenter the minorities in the narrative (Fraser 1991).

In contrast, activism as connective action is sustained by emotional and affective engagement (Papacharissi 2016) recognized and confirmed continually through social media affordances such as 'likes' and 'favorites' (Milan 2015, pp. 62–63). Public communities are often based on a personal vulnerability or empowerment and are performed through individual testimonies expressed and disseminated on social media platforms but in a way that creates a feeling of community and solidarity. The understanding of what is common to the community is redefined and creates a "transposed identity building", which Milan (2015, p. 62) terms "visibility". The virtual body exists by virtue of its online and public presence and recognition of the personal vulnerability as an individual part of a new kind of collective. This effect of mediatization produces the body as a sign embedded in affective politics (Reestorff 2014, p. 489). Bennett and Segerberg finally theorize a hybrid connective action which produces affective publics organized both off- and online. In this hybrid connective action, the activist body is constructed digitally in a way that is autonomous while also embedded in non-hierarchical, polycentric, and digitally supported organizational and political structures.

Digital technologies, however, may also "enable hybridity, diaspora and cosmopolitan affiliations" (Ponzanesi 2020, p. 978). Postcolonial critique presents geopolitical and historical specificities as important perspectives that challenge and change the concepts of counterpublics and connective action by insisting on the existence and significance of embodied cosmopolitan and postcolonial bridges (Blaagaard 2018). What theories of social media activism are lacking, then, are the strategies of empowerment and vernacular expressions found in empirical examples of decolonial activism. Being present in the digital collective is not enough. We need ways of identifying pathways from the individual, affective, and political body to the social movement.

## 4. The Postcolonial Activist

I do not want to dismiss the importance of the role of technology, and, in particular, the recent new digital media technologies, in formulating and supporting activism which empowers minorities and political counterpublics. Arguably, digital media have enabled enclaved publics that rely on scripted interaction with the dominant public and "'hidden transcripts' in safe spaces" (Squires 2002, p. 458) to become counterpublics with the resources and audiences to engage in opposition to the dominant public's narrative (Squires 2002, p. 460). Nevertheless, the technology-centered theories rehearsed above seem to consign the embodied activism to the perceived outdated practice of collective action such as sit-ins and marches and the digital connectivity activism to a disembodied individualism and visibility doctrine ripe for exploitation. Scholars of postcolonial critique and critical race theory, however, have produced counterarguments and theorized postcolonial activism differently. In the following sections, I want to discuss and present five conceptual practices, i.e., practices and positionings that allow us to think through the postcolonial potential and relations in activism (Blaagaard 2018, pp. 12–13). The practices and positionings function as navigational tools through which I analyze BLM-DK's digital and postcolonial counterpublic within and beyond the technological narrative.

### 4.1. Practices . . .

If collective action is associated with marches and sit-ins and connective action with visibility and individual experiences shared online, *vernacular expressions* is a practice that points to a subjective expression which, in turn, assembles groups of like-minded individuals.

Gilroy (1993) theorizes Black vernacular expressions in the music of African American and African Caribbean diasporic communities. Music draws on cultural and collective memory, as well as the contemporary and political situation in which the practitioner finds herself. As such, it is "embodied practices of remembering colonial culture and continuing postcolonial struggles through art" (Blaagaard 2020, p. 313). Gilroy's vernacular expressions call on countercultures and counterpublics and produce them through the circulation and repetition of vernacular knowledge produced in subjects' everyday lives and in resistance to dominant cultural narratives (Fraser 1991; Warner 2002). In new digital media, visuality is gaining attention. Other vernacular expressions may be visual, such as *embodied hashtags* (Blaagaard 2022) or "corporeal iconography" (Richardson 2020, p. 143), that bind an embodied expression to visual and digital imagery or a sign, which, in turn, conjure up a cultural memory of a shared struggle and past. An example of these expressions is the #IRunWithMaud running activism, which campaigned to support the judicial case against the killers of Ahmaud Arbery, who was killed while exercising. The campaigners started their campaign on Arbery's birthday (8 May 1994) by running 2.23 miles, signifying the date of his death, 23 February, and continue to share on social media their technological fitness devices displaying the miles run (see also Blaagaard 2022). When the physical act of running is coupled with viral technology, it becomes an embodied hashtag—i.e., the ordering technology which connects the embodied and visual actions. Vernacular expressions of music and imagery used online and infused with the viral abilities of networked connectivity reappropriate the technological propensity for visibility of the mediatized body by connecting it inseparably to the embodiedness of the

physical experience; i.e., singing and playing music and making and repeating movements and gestures are inseparable from the singular, physical body.

While the dangers of appropriation and exploitation are plenty when it comes to digital creative expressions of resistance, Steele (2021) argues that "Black women have long found chasms within which to undermine the logic of this system of oppression and craft space to survive and thrive" (p. 34). Turning our attention to vernacular expressions as tactical devices for social change, we find rhizomatic structures of resistance, which Steele calls *survival technologies*. Gilroy (1993) also heeds the complex relationship between political, postcolonial resistance and commercialization and commodification. Postcolonial creative expressions are cultural products that interact with and help reshape the parameters for esthetic appreciation, always within a discursive power relation (Ponzanesi 2014).

Embodied hashtags reinterpret technology creatively. What was once a device for measuring athletic progress is now a political act of endurance. Fouché (2006) writes that Black vernacular technological creativity is "the innovative engagements with technology based upon black aesthetics" (p. 641). The creativity spans a continuum from weaker to stronger expressions. Fouché presents the Black vernacular technological creativity as redeployment in which technologies are reinterpreted and used beyond their intended or imagined purpose. This is a critical approach to technology that, through reappropriations and mash-ups, rethinks technology's abilities. Reconception of Black vernacular technological creativity is expressed as a usage which subverts the purpose of the technology. I characterize this approach as a deconstruction of technology because it shows the underlying abilities of the product while also reevaluating its potential. Finally, Fouché theorizes Black vernacular technological creativity as re-creation, as a practice which brings about new material inventions on the basis of older and discarded technologies (p. 642). This is the decolonization move in Fouché's theory. In recreating technology, Black vernacular technological creativity not only reinvents or reevaluates, but also fashions another and independently positioned technology. If we are to understand Fouché's use of the term technology in Foucauldian terms, Black vernacular technological creativity is broadened to encompass the everyday practices of African Americans (Fouché 2006, p. 640; Steele 2021, p. 32). Technology is no longer defined as an object, rather "[t]he move away from the object, to the person or the community, creates new opportunities to study the ways those marginalized engage technology within their everyday lives" (Fouché 2006, p. 650). Technology is the practices and positionings of and within communities.

### 4.2. . . . and Positionings

Recreation brings a need for a reflexive and creative positioning that is delinked from the Western epistemology and dominant culture. This positioning is *border thinking* (Mignolo 2013). The borders in Walther Mignolo's border thinking theory are not geographical but epistemic. Mignolo asserts the necessity of decolonizing the Western epistemologies and insisting on committing epistemic disobedience (pp. 136–37). Akin to Fouché's recreation of technology, Mignolo's border thinking takes up an independent position complete with its own genealogy, cultural memory, and knowledge. Border thinking provides a critical memory created through "a continuity in the development of black publicity rather than [ ] recurrent novelt[ies]" (Baker 1994, p. 15). Similarly, in her introduction to *Digital Black Feminism*, Steele (2021) places "Black women at the center of conceptualizing technology and digital culture. [ . . . ] Black women's historical and persistent relationship with technology provides the most generative means of studying the possibilities and constraints of our ever-changing digital world" (p. 1). While critical, cultural memory could reasonably be theorized as counterpublicity, Brouwer and Paulesc (2017) remind us that the counterpublic is always referring to a situation or dominant public to which it reacts, making it dependent on the dominant culture's existence (p. 87). Counterpublic theory therefore runs the risk of arguing for the inclusion of politically marginalized minorities rather than the deconstruction or even decolonization of the structures of the narratives holding up the binary of 'here' and 'there', 'then' and 'now', 'home' and 'abroad'. The

counterpublic wants to be included in *the* public, 'here', 'then', and 'home'. This is not the case for vernacular expressions, be they technological or not. "[V]ernacular expressions care less about being heard by Western media than about enacting a political community and culture in and of itself" (Blaagaard 2020, p. 314).

Genealogies of vernacular expressions may, moreover, produce and be produced by *cosmopolitan bridges*. Blaagaard (2018) discusses the concept of cosmopolitan bridges through a critique of the perceived objectivity of technology. While technology is seen as neutral, it is rather the opposite (Blaagaard 2018, p. 58; Fouché 2006, pp. 648–51; González and Torres 2011). Technological advances from the wire to the camera are imbued with racial implications. So, for example, while journalism was believed to spread factual stories through the wire at the turn of the century, thereby creating a common understanding and vision of the world, it was, in fact, excluding countless communities by using stereotypes and reductive representations to fit the short format of the telegraphic messages. In contrast to this technologically focused objectivity and derived cosmopolitanism, Blaagaard argues with Rantanen (2003) that cosmopolitanism must rather be understood as the practice of connecting discrete but embodied places, thus producing bridges that rework relationships through creative practices (Blaagaard 2018, p. 60). The example presented by Blaagaard is the practice of the postcolonial newspaper *The Herald*, which was produced between 1915[3] and 1925 in the Danish colonies of the Danish West Indies, the present-day United States Virgin Islands. The newspaper was filled with articles reproduced from political progressives in Denmark and civil rights champions in New York, through which cultural memories and common experiences were produced and shared. Cosmopolitan bridges not only connect discrete places and people, but also minds and ideas. In this way, the Danish colonial newspaper is part of the continuity of Black publicity.

To sum up, the postcolonial activist is made up of the embodied and genealogical practices of *vernacular expressions*, *embodied hashtags*, and *survival technology* and the positionings of *border thinking*, and *cosmopolitan bridges*, which are supported by but cannot be reduced to the connectivity of technology. Because these practices and positionings are inherently embodied, counterpublics emerging from them do more than oppose a dominant public. They create a genealogy, a cultural memory or a changing same (Gilroy 1993) of discrete spaces and bodies that invites rhizomatic relations between situated and political events across geographical space and historical time. In the case study of Black Lives Matter Denmark, I explore this kind of counterpublic further.

## 5. The Positionings and Bridges of Black Lives Matter Denmark (BLM-DK)

BLM-DK's Facebook page was created on 25 September 2016. At the time of sampling, the community page is just short of 18,000 followers. The page operates as a chapter of the international organization of BLM, which was founded by Alicia Garza, Patrisse Cullors, and Opal Tometi, who circulated the hashtag #BlackLivesMatter after the acquittal of Trayvon Martin's murderer in 2013 (Richardson 2018, p. 388). However, the link between the US and DK chapters could be said to be connective and affective rather than collective in nature. For instance, the core issue of the international BLM organization is the reformation of the US police force to stop the many incidents of killings of Black people by police. The killings are often followed by impunity and indifference in US political society, leading to and sustaining the prison complex. Although the Danish and the US police forces' structure and reach are very different, the issue of US police violence is forwarded on BLM-DK's Facebook page.

BLM-DK shared a Twitter post by Kwame Rose and @breatheofficialpage (20 March 2022) that told the tragic story of how another young Black man was killed by police in Baltimore only six years after he featured in a *Time Magazine* article discussing the impact of the death of Freddie Gray by the hands of the police in the very same city. The connecting timelines and events emphasize the continuity of the killings, the circulation of racial profiling and impunity always repeating itself. The narrative was carried over to a wider circle and context when BLM-DK shared a news story (28 March 2022) about a young

Danish man of Middle Eastern descent who found himself being stopped by the police when driving. The man argued that his ethnicity plays a role in the continued interest shown in him and his movements by the police. The article is a follow up to a report published by the Institute for Human Rights, showing that the practice of racial profiling is widespread in Denmark (9 April 2022). Similarly, in London, a child of African and Caribbean descent was strip-searched at school by police after being wrongly accused of carrying drugs. The incident brought Londoners to the streets in protest, pictures and news stories of which @antoinespeaker covered and BLM-DK shared (20 March 2022). The sharing of these tweets and posts by the Danish BLM chapter brings the events in the US, DK, and UK into a common context of police violence and impunity, allowing these stories from separate spaces to share a platform and construct a connected narrative and creating a cosmopolitan bridge between the geopolitical spaces. The cosmopolitan bridges support a counterpublic, which is not global but transnational and postcolonial. They enable common ground as well as differences across the Danish, UK, and US police forces and reach.

While these bridges clearly show that racism is transnational and that Black and Brown people everywhere share an experience of being profiled and targeted, they also show the global differences. The Danish Middle Eastern man who was targeted by the police shared his experience with Black people in the US and a child in London, but they are not the same because of the differences in history and politics. The "romantic sugary narrative of how we hid and protected 'Our Jews' during the occupation [during WWII] is becoming more and more untenable", writes BLM-DK as a comment on the article about the profiling of the Danish Middle Eastern man (28 March 2022). Thus, the issue of police profiling is connected to a particular Danish national identity and used to argue for social and political change in a particular national and postcolonial setting. Juxtaposing the experience of racial profiling today with the national(istic) narrative of Danish behavior during the Nazi occupation between 1940 and 1945 deconstructs Danish national identity and the Danish relationship to racialized others using national history as well as contemporary but cosmopolitan knowledge.

Another bridge constructed on the BLM-DK Facebook page concerns the experiences of racism at the border between Ukraine and Poland. Earthrise (18 March 2022), which is the name of activist campaigner Alice Aedy's website, on which she displays her activism in photos and documentaries, NBC (6 March 2022), and Good Morning Britain on British ITV (7 April 2022) all posted stories about issues that were shared by BLM-DK. The stories of Black refugees leaving Ukraine because of the Russian invasion throw into relief the difference between the reception of White and Black people fleeing Ukraine. While White Ukrainians were welcomed in Denmark, Black people were detained at the Polish borders. The story was amplified and supported by local discussions posted by BLM-DK about the Danish government's inhumane migration policies (15 March 2022) and general racism in Danish society (9 April 2022). Again, the collective indignation creates a transnational counterpublic—a collective space set against the dominant public. However, in contrast to police violence, which connects to a US context and genealogy, arguably the issue here is specifically postcolonial and European. Crossing borders and drawing new boundaries is the plotline of colonialism. Using Etienne Balibar's political concept, expertly discussed by Stoler (2020), the posts dealing with the plight of Black and Brown Ukrainian refugees produce and reflect *interior frontiers* "whose fluctuating parameters mark diffracted histories of the present" (p. 120). The interior frontiers show the constantly negotiated corridors through which racialized others and the othered travel and make up postcolonial Europe's map and borders.

Understanding the cosmopolitan bridges in relation to these interior frontiers, it is clear that the bridges do not produce a unified counterpublic but a hybrid space of postcolonial connectivities. Interior frontiers perceive and understand these bridges and cosmopolitanism from the inside, from below, embodied and always in flux and negotiating space and boundaries. In this way, BLM-DK becomes a part of the continuity of a postcolonial publicity rather than a unified counterpublic.

## 6. Postcolonial and Vernacular Practices

The practices displayed on the BLM-DK Facebook page underscore the bridges and emphasize the creativity at play on social media platforms and communication. In the following, I focus on four individual examples of visual–linguistic practices: a visual mash-up using an image of a tweet from a Danish conservative politician, Marcus Knuth, next to a pair of socks emblazoned with the words "Shut up" (5 March 2022); an invitation to celebrate the passing of the CROWN Act in the US House of Representatives using an image of a Black woman with a natural hairstyle and asking followers to send images of their own natural hair (21 March 2022) and an image of a child squatting in front of a sculpture of a woman in shackles throwing herself forward and towards the child (23 March 2022). Their eyes seem to meet, and they hold each other's gaze. The image is taken at the National Memorial for Peace and Justice, Birmingham, Alabama; and finally, citizen journalistic sousveillance practices as they pertain to the events at the Ukrainian border (4 March 2022; 6 March 2022) and to an incident in which an asylum seeker from Iran was forced to leave her child and husband to be sent back under the new Danish immigration policies (29 March 2022).

### 6.1. Laughter

Connected to the posts on the difficulties facing Black and Brown refugees at the Polish border, the mash-up of Marcus Knuth's tweet and the "Shut up" sock is accompanied by the hashtags #BLM, #BlackLivesMatter, #StandWithAllRefugees, and #EnFlygtningErEnFlygtning (a refugee is a refugee). The two latter hashtags explain the standpoint of the two former hashtags in that they argue for equality among refugees and for the standpoint that the color of a refugee's skin does not determine their human value and need for help. Paraphrased, Knuth's tweet says that it is "obvious" that Denmark will help the Ukrainian refugees to safety but that "we say no to inviting 2.300 Afghans and Syrians . . . and potentially 10.000 more from the Middle East". The Afghans, Syrians, and Middle Eastern people of whom he speaks are people who had been granted entry into or even asylum in Ukraine and who are now fleeing war and unrest once again. His tweet is adorned with an image documenting how many successful asylum seekers there are in Ukraine. The accompanying photo of the "Shut up" sock may be self-explanatory, drawing on the expression 'stuff a sock in it!', meaning 'shut up'. In this way, BLM-DK make their opinion clear in a humorous manner. The mash-up is an example of Black vernacular technological creativity as redeployment (Fouché 2006), in which technologies are reinterpreted and used beyond their intended or imagined purpose through a critical reappropriation. In this case, the reappropriation asserts an opinion which discursively circulates to produce a counterpublic. While the singular post enables a counterpublic, it is its cosmopolitan bridges and interior frontiers that inscribe it into the postcolonial genealogy and publicity.

It is also worth noting that, like singing and gesturing, laughing is an embodied process. An embodied hashtag, then, may be produced through mash-up posts.

### 6.2. Representation and Recognition

Now I turn to the two posts that respectively feature a call to celebrate natural hair as the CROWN Act is passed in the US House of Representatives and an image from the Alabama national memorial. The former refers to the House of Representatives as simply "the House", which demands knowledge of the US political system and what is being discussed there. The post also refers to the Act without grounding it in the US political context, merely commenting that it is "ridiculous" that the issue of discrimination against African hairstyles was "even up for discussion". The text moreover explains the meaning of the acronym CROWN. The lack of explanation and context paired with the unanswered call for followers to send in pictures of their natural hair leaves the post in a strange position. On the one hand, it speaks to Black people everywhere who have experienced discrimination due to their hairstyle. On the other hand, the particular political and legal Act, the CROWN Act, which is celebrated is not relevant to the Danish followers or indeed



to anyone outside the US. Similarly, the post from the National Memorial for Peace and Justice in Birmingham, Alabama, is uprooted and yet found relevant by the followers of BLM-DK. The image is headed with the text in red letters: "From Black women before me, I draw endurance, tenacity, perseverance [sic] but most of all . . . love & strength". The post is not referenced, and it is unclear who is uttering the words, who the girl in the picture is, and when the image was taken and by whom[4]. Yet it produces an almost universal and temporal claim to humanity in the interlocking gaze between the representation of the past (sculpture of the woman in shackles) and the flesh and blood of the future (the squatting girl). The image shows an embodiedness of temporal, spatial, and material relations.

The two posts, then, are undeniably American. They are rooted in US history and politics while also calling on the global community of Black experience when posted on BLM-DK's page. The images produce a counterpublic less postcolonial and critical than the cosmopolitan bridges discussed above. However, they are examples of creativity and digital expressions pushing back against the mainstream media representation. The embodiedness of both hair and gaze connects social media followers in a human experience defined by African perspectives rather than European history. Their positions echo Steele (2021, p. 1), who "place[s] Black women at the center of conceptualizing technology and digital culture. [ . . . ] Black women's historical and persistent relationship with technology provides the most generative means of studying the possibilities and constraints of our ever-changing digital world".

*6.3. Authenticity of Esthetics*

Twice during the two months of sampling, BLM-DK shared or produced sousveillance using mobile devices. Richardson (2020) argues that the mobile device witnessing practiced among Black people to document excessive police force and violence is both a community-producing act as well as a call to set the record straight. The acts of witnessing have made available to all the experiences of Black people in the US. It could also be termed survival technology. BLM-DK's spokesperson Bwalya Sørensen traveled to the Poland–Ukraine border (4 March 2022; 6 March 2022) to help document how Black and Brown people are treated. The first post shows Sørensen at the train station in Przemysl sharing her findings in selfie-style reportage (4 March 2022). The second post shows the area around the soup kitchen where activists have set up a piano "soothing everyone with music" (6 March 2022). While these posts do not expose maltreatment or deception, they carry an authenticity in the esthetics that brings the body of the refugees into focus.

A third post using mobile device witness situations from a perspective from below is a film shared by the refugee help center Trampoline House and shared by BLM-DK (29 March 2022). This film shows the violent detention of a Kurdish Iranian mother who has had her application for asylum rejected and is being forcefully separated from her husband and her baby to be taken to the airport and sent to Iran. The Danish government is implementing regulations stating that refugees can be sent back to areas such as Syria and Iran. Denmark is the only country in Europe that finds Syria safe, and often it is only some members of the family who are deemed safe to return. Needless to say, this practice breaks up families and sends people back to violent and unsafe locations. This is what happened to the Kurdish Iranian woman. She is seen being flung to the ground by several detention officers and allegedly given a sedative.

The film shows many people filming the incident, and the procedure and the woman's fate became news in the mainstream media[5]. The film serves to set the record straight and to document the indecency of the Danish refugee system. Using Fouché's terminology, mobile device footage such as this is a redeployment of technology. These films are neither humorous nor situated knowledges producing universal epistemologies from below. Rather than the beautifying and filtered esthetics of social media platforms, the esthetics of citizen journalism are those of a different embodiedness, one that draws on authenticity and documentation rather than representation. The grainy and shaken quality of the imagery is a particular "technology-expressed aesthetics of what is happening" (Blaagaard 2015,



p. 58), an esthetic generated and sustained by our lives in media. The embodiedness of these imageries relies on a viewer who is willing to suspend her vision and believe that reality in chaos and crisis is blurred, grainy, and shaken, which, of course, human vision is not. In different ways, then, the postcolonial practices discussed in this segment all draw on the embodied experience through technology: through laughter as an embodied hashtag, through representation and recognition allowing a transnational exposure and value, and through the authenticity of esthetics as technological vision.

## 7. Concluding Discussion

To what extent may BLM-DK be said to produce a counterpublic independent of the mainstream and dominant public? And to what extent are rhizomatic structures discernible in the sample? While the connected fight against police brutality, the celebration of Black hairstyles, and the sousveillance that documents the oppression of Black and Brown people and sets the record straight all produce a counterpublic against a White system of political and cultural dominance; the counterposition in all these cases implies the dominant public. In this case, as with other cases of counterpublicity, BLM-DK reproduces the dominant public's importance and power by relying on its existence. However, the technological affordances of social media and the cosmopolitan bridges they support immediately hybridize and diversify the counterpublic position. Excessive police force is used to deconstruct the Danish identity, and sousveillance is used to document the inhumane and destructive migrant policies in Denmark rather than the shooting of Black youth. The grounding of counterpublicity in distinct national and political contexts challenges the unified cause of the counterpublic while simultaneously producing the connectivity that helps build the political power of postcolonial and critical race issues. This structure of activism could be akin to what Bennett and Segerberg call the hybrid form of connectivity; but it is not the structure which is of interest here. The embodiedness of the counterpublic's members is not necessarily the collective factor but is rather produced transnationally and rhizomatically in diverse contexts. The postcolonial shifting boundaries of the interior frontiers that develop and the technological vision which is supplied point towards an emerging postcolonial survival technology, supporting bridges between bodies.

Postcolonial counterpublics must be understood through singular cases so as not to reinstate a homogenous counterposition between a norm and a racialized othered postcolonial community. BLM-DK is particular in its cultural and political work, connected to a Westernized (Americanized) focus as well as a European and Danish political frame, and a strange, askew perspective is produced; while the cosmopolitan bridges provide vantage points from which to encounter the transnational likenesses across the Atlantic, the case study underscores that Danish politics concerning refugees and migrants are different from those in the US, and European history is different from that of the US. And so, while BLM-DK connects to civil rights history and African American history, they do not explicitly draw on the Danish history of enslavement and slave trade in their mediated communication. This exclusion of Danish history as a power relying on slavery, and the common history of Denmark and the US in the US Virgin Islands, in the narrative of the BLM-DK's Facebook page perhaps signifies a slant in the sample or in the narrative of BLM-DK.

Nevertheless, in the case of BLM-DK, the social media platform becomes a platform for *survival technologies* to produce *embodied hashtags* in the shape of sousveillance esthetics and epistemologies from below. The *vernacular expressions* identified in the sample show emerging *cosmopolitan bridges* and rhizomatic structures that challenge the uniform counterpublic. While social media may be blind to the racial history and politics behind technology and may, in this way, replicate racial structures and idioms such as ideas of Black technophobia, simultaneously, counterpublics of political resistance use strategies of vernacular expressions and social media savvy to underscore and recenter their experiences. The particular postcolonial vernacular is the process of bridging spaces of subjugation and, in turn, empowering the political impact: not a case of producing relations of either–or, or

even both–and (them–us), but rather an instance of producing a rhizome of additions, of and–and–and (see Grossberg 2002, p. 369) beyond the technological imperative.

**Funding:** This research received no external funding.

**Institutional Review Board Statement:** Not applicable.

**Informed Consent Statement:** Not applicable.

**Conflicts of Interest:** The author declares no conflict of interest.

## Notes

[1] The exact number of different outlets is unclear as some of the posts have no clear sender.

[2] The section on the activist of new media is unfolded and discussed further in (Blaagaard and Roslyng 2022).

[3] The islands were sold and transferred to US jurisdiction in 1917. The inhabitants of the islands were granted US citizenship ten years later, 1927. This period in the islands' history therefore presents us with complex relationships and negotiations between identity, citizenship, and rights claims (Blaagaard 2018).

[4] The location of the sculpture was ascertained using Google image reverse search.

[5] See (Roslyng and Blaagaard 2022) for a discussion of how citizen journalism circulates.

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
