# Peer review of "The Practices and Positionings of a Postcolonial Counterpublic: An Analysis of Black Lives Matter in Denmark"

_humanities, doi:10.3390/h12040061_

Round 1

Author Response

Thank you for thorough and engaged comments and suggestions. I have addressed them all in the attached document.

Reviewer 2 Report

This is an interesting area of study and overall I enjoyed reading the manuscript. I have stated that the scholarship is average and satisfactory because I would have liked a better explanation regarding the aim of the paper. The writing is complicated in a number of places, unnecessarily so - we are after all in the business of writing accessible text and this is particularly so when we are discussing social movements such as #BLM. Furthermore, there needs to be a better and clear link to the postcolonial critique. The research has been taken from facebook over a short period of time with a small sample and although the author states it, there needs to be a better explanation why this was done in 2022. Good use of examples. It would have been useful to note how the Danish chapter of #BLM came into existence. The structure of the article needs attention to make it clear about the link between collective and connective action featured in social movement theory. The authenticity of aesthetics section was relevant and this should really be connected to the 'research' which is done using social media ie how much of the story is really authentic? This is also related to the ethics of doing this sort of research - do we have permission to write about people? Was there any attempt to contact the #BLM chapter? Overuse of Blaagaard texts. References should be in alphabetical order.

Author Response

Thank you for your thorough comments and suggestions. I have addrssed them all in the attached document.
